# Prevalence of Intestinal Parasites in HIV/AIDS-Infected Patients Attending Clinics in Selected Areas of the Eastern Cape

Anozie Ifeoma [1], Teke Apalata [2], Boyisi Aviwe [3], Olanrewaju Oladimeji [1] and Dominic T. Abaver [2,3,*]

[1] Department of Public Health, Faculty of Health Sciences, Walter Sisulu University, Nelson Mandela Drive, Mthatha 5117, South Africa
[2] Division of Medical Microbiology, Department of Laboratory Medicine and Pathology, Faculty of Health Sciences, Walter Sisulu University, Mthatha 5117, South Africa
[3] HERENDA Program, Department of Laboratory Medicine and Pathology, Faculty of Health Sciences, Walter Sisulu University, Mthatha 5117, South Africa
* Correspondence: dominicabaver@yahoo.com

**Abstract:** Introduction: Intestinal parasites in HIV and AIDS patients increase the risk of gastroenteritis, adding to the complexity of the virus. According to the literature, their interactions are one of the factors leading to HIV replication and progression of AIDS in Africa. Chronic immunosuppression caused by HIV infection makes people vulnerable to parasitic infections, and this is associated with a CD4+ cell count of less than 100. The study describes the prevalence of intestinal parasites in patients attending HIV/AIDS clinics in certain areas of the Eastern Cape. Methods: A cross-sectional study was conducted among 600 patients from HIV/AIDS clinics in the Eastern Cape. Tambo Municipality and Amatole Municipality were the municipalities covered. These included the Ngangalizwe Community Clinic, Tsolo Gateway Clinic, Idutywa Health Centre, and Nqamakwe Health Centre. The stools of 600 participants were examined using direct wet saline/iodine embedding, formal ether concentration technique, and modified Ziehl–Neelsen methods. Results: The mean age of the study participants was 28.2 years. They were predominantly female (79.9%), mostly single (63.6%), and lived in rural (65.2%) and urban areas (34.8%). The prevalence of intestinal parasites was determined to be 30% (180/600) after screening 600 stool samples. The most frequently detected parasites were *Ascaris lumbricoides* (55.9%), *Balantidium coli* (15.1%), *Entamoeba* coli (11.3%), *Diphyllobothrium latum* (4.3%), *Taenia* species (3.8%), *Schistosoma mansoni* (1.6%), and *Cryptosporidium* spp. (1.6%). Males were affected more frequently (39.2%) than females (27.9%). The difference was statistically significant ($p = 0.017$). Among the identified intestinal parasites, *A. lumbricoides*, *B. coli*, and *Taenia* spp. were found at all four sites. Conclusion: This study has shed light on the high burden of intestinal parasites in HIV/AIDS patients in the Eastern Cape. Medication adherence, deworming, and sanitary hygiene practices are needed to enhance the control of infection in the affected communities and hence contribute to the control of the HIV pandemic.

**Keywords:** HIV/AIDS clinics; intestinal parasites; prevalence

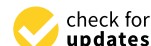



## 1. Introduction

The global distribution of intestinal parasites, as well as their impact on morbidity and mortality, is well documented [1,2]. Up to 60% of the world's population is infected with intestinal parasites, which has a negative impact on morbidity rates, particularly in Sub-Saharan Africa, where 70% of HIV/AIDS infections occur [3]. HIV and parasitic infections in the intestine interact cause rapid HIV spread and progression from asymptomatic HIV infection to AIDS [4]. HIV infection causes chronic immunosuppression, which predisposes to parasitic infections and is associated with a CD4+ cell count of less than 100 cells [5].

Parasites, such as *Cyclospora* spp., blastocysts, *Schistosoma mansoni*, *Isospora belli*, microsporidia (e.g., *Enterocytozoan bieneusi* and *Encephalitozoon* spp.), Giardia lamblia, *E. coli*, and *A. lumbricodes*, have been identified in HIV/AIDS patients [6]. Intestinal parasites, such

as Schistosoma mansoni, represent a clear and potential risk factor for the transmission of HIV-1/AIDS through intrahost interactions leading to disease progression [7]. It has been suggested that HIV-positive people should be screened for intestinal parasites as they are prone to coinfections [8]. In a cross-sectional study conducted in Kenya, HIV/AIDS patients had a prevalence rate of 50.8% [9]. A comparable study in Burkina Faso, West Africa, found a prevalence rate of 24.73% [10]. The results of the study among adult HIV patients in South Africa, where most HIV/AIDS cases are concentrated, showed a helminth prevalence rate of 36.1% [11]. The Eastern Cape study found a 25% prevalence of intestinal parasites in HIV/AIDS adults [12]. The purpose of this study was to determine the status of intestinal parasites in patients attending HIV/AIDS clinics to enable evidence-based intervention.

## 2. Materials and Methods

### 2.1. Study Setting

This study was a cross-sectional design conducted in four clinics located in two districts in the Eastern Cape, South Africa. The four clinics, which were conveniently selected, are the Ngangalizwe Clinic, Tsolo Gateway Clinic, Idutywa Health Centre, and Nqamakwe Health Centre [13].

### 2.2. Study Participants

The study participants were all HIV/AIDS clinic-attending patients. Pregnant women and people who were on antiparasitic treatment at the time of the study were excluded. A total of 600 HIV/AIDS outpatients participants were recruited and examined.

### 2.3. Data Collection and Laboratory Investigation Procedures

Stool Collection and Examination

Clean and dry leak-proof containers with lab identification number were distributed to the participants. They were instructed to pick no less than 5 g of the stool specimen. With the provision of stool specimen, a structured questionnaire was administered to determine sociodemographic characteristics, individual level of knowledge, attitude, and practices that were risk factors of intestinal parasites. Proper documentation of patient was done.

The diagnosis of intestinal parasites was made with positive microscopic stool specimen samples. The stool was examined macroscopically and microscopically. The macroscopic examination checked for consistency, colour, blood stains (diarrhea), and worms or their segments while the microscopic examination checked for the presence of trophozoites, cysts, oocysts, larvae, and ova of intestinal parasites. Each specimen was examined using the direct wet mount method using normal saline (0.85% NaCl solution), concentration method, and Ziehl–Neelsen method at the microbiology lab of Walter Sisulu University at Mthatha Campus).

Microscopic examination of stool sample was done by applying saline. The sample was applied to a small area on a clean microscope slide. Big fibres and particles were removed. One or two drops of saline were added using a pipette. The specimen was mixed by allowing bubble creation, and it examined under low-power objective $10\times$ and low light. Thereafter, it was closely examined to identify parasite-like objects and species under microscope. Saline wet mount is useful to detect live motile trophozoites of *E. histolytica*, *Giardia lamblia*, and *Balantidium coli* and cysts of protozoa and eggs and larvae of helminthes.

Modified acid-fast staining for coccidian oocysts (Ziehl–Neelsen staining) was also applied to detect protozoa, such as oocysts of *Cryptosporidium*, *Isospora*, and *Cyclospora cayentanensis*. The duration of the study was from March 2019 to February 2020. The samples from the four sites were collected and analyzed between March 2019 and February 2020.

### 2.4. Data Processing and Analysis

Data were analyzed using SPSS version 27.0 software (Walter Sisulu University, Umtata, Eastern Cape, South Africa). The prevalence of intestinal parasites in the sites covered was determined. The prevalence of intestinal parasites in relation to different sites and

variables was determined by Chi-square, which was used to assess significance difference. A statistical test result was considered significant if the *p* value was less than 0.05.

### 2.5. Ethical Consideration

The study was approved by the faculty of health sciences postgraduate education, training, research and ethics unit, and human research committee of Walter Sisulu University (protocol number 106/2018). The department of health and local municipalities where the studies were carried out gave permission for the study to be carried out in their clinics, and the clinics gave permission for the team to have access to their patients to be enrolled as participants. Informed consent was obtained from all adults who were 18 years and above. The participants who were below 18 years were consented to by their parents or guardians. Those patients found positive with parasites were treated using the standard drugs approved by WHO anti parasitic drugs.

## 3. Results

### 3.1. Sociodemographic Characteristics of Study Participants

The sociodemographic characteristics of all the study participants at the four selected sites in the Eastern Cape region, South Africa, 2020 are in Table 1. The valid percentages presented exclude the missing values.

**Table 1.** Sociodemographic characteristics of all research participants at the four selected sites in the Eastern Cape (*n* = 600).

| Variable | Characteristic | Frequency | Percentage |
|---|---|---|---|
| Gender | Male | 120 | 20.1 |
| | Female | 476 | 79.9 |
| Age (years) | 12–20 | 33 | 5.5 |
| | 21–30 | 106 | 17.7 |
| | 31–40 | 181 | 30.3 |
| | 41–50 | 162 | 27.1 |
| | 51–60 | 78 | 13.0 |
| | 61–70 | 32 | 5.4 |
| | 71+ | 6 | 1.0 |
| Place of residence | Rural | 391 | 65.2 |
| | Urban | 209 | 34.8 |
| Number of toilets in the house | 1 | 500 | 83.3 |
| | 2 | 89 | 14.8 |
| | 3 | 5 | 0.8 |
| | No toilet | 6 | 1.0 |
| Type of toilet used | Traditional | 404 | 67.3 |
| | Modern style | 171 | 28.5 |
| | Both | 20 | 3.3 |
| | Bush type | 5 | 0.8 |

As shown in Table 1, 600 participants were included in the analysis of this study. Of them, about 476 (79.9%) were females and 120 (20.1%) males. Those within the ages of 31–40 were proportionally higher 181 (30.3%), followed by those within the ages of 41–50 (27.1%). More than half of the participants, 381 (63.6%), were single, and 186 (31.1%) were married. Those who have completed high school were more with 62.4%, followed by those who have completed primary education, 165 (27.7%). Those residing in rural areas were 391 (65.2%) and in urban areas 209 (34.8%), with family sizes mainly with a size of 3–4 members (41.67%) and greater than 4 members (41.5%). The number of rooms used was more within 3–4 250 (41.67%), and the number of toilets was higher in those that said just one, 500 (83.3%), and the type of toilet was primarily traditional, 404 (67.3%).

*3.2. Isolation and Distribution of Intestinal Parasites at the Four Different Study Sites*

The total and types of different intestinal parasites isolated are shown in Figure 1 below. The prevalence of intestinal parasites among study participants was 30% (180/600)**.** Fourteen parasites were detected with a dominance of helminths (70.4%) over protozoa (29.6%). The parasites in the study were *A. lumbricoides* 55.9%, *B. coli* 15.1%, *E. coli* 11.3%, Diphyllobothrium *latum* 4.3%, *Taenia species* 3.8%, *Enterobius vermicularis* 3.8%, *Schistosoma mansoni* 2.1%, *Cryptosporidium* spp. 2.0%, *Fasciolopsis busiki* 1.1%, *Trichuris trichiuria* 1.1%, *Isospera belli* 0.5%, *Hymenolepsis nana* 1.6%, *Fasciola hepatica* 0.5%, and *Trichostrongylus specie* 0.5%. The opportunistic parasites found among them were *cryptosporidium* spp., *E. coli*, *Balantidium B. coli*, *Isospera belli*, and *Enterobius vermiculari*, and they accounted for 30.5%. The most detected helminth was *Ascaris* (55.9%), and the most detected protozoan was *Balantidium coli* (15.1%).

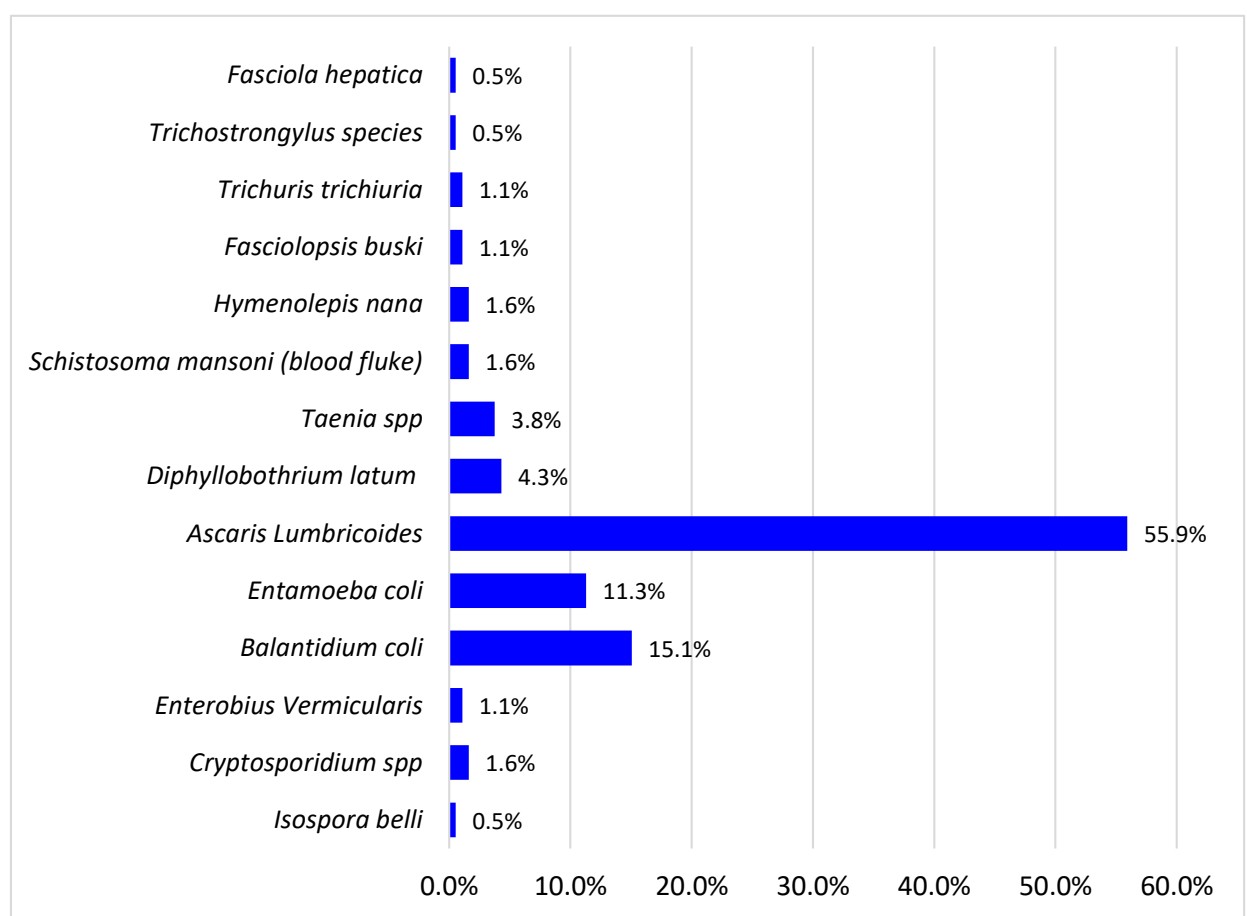

**Figure 1.** Intestinal parasites isolated.

3.2.1. Site Prevalence

The site with the highest prevalence of intestinal parasites was the Nqamakwe Health Centre at 44.3% (47/106). The site with the lowest prevalence of intestinal parasites was the Tsolo Gateway Clinic, 20.8% (30/144) as shown in Table 2. The majority of the HIV/AIDS participants from Nqamakwe live in the rural areas. The Chi-square test for independence was used to determine if a relationship existed between the clinic and the presence of parasite. The results show a $\chi^2 = 17.999$ (*p*-value = 0.001), implying a significant relationship between the clinic and the presence of intestinal parasites. In other words, a higher prevalence of parasites was statistically significantly observed in the Nqamakwe Clinic, followed by the Ngangalizwe Clinic.

**Table 2.** Prevalence of intestinal parasites by site (clinic).

| Clinic | Intestinal Parasites | | Total | Df | Chi-Square ($\chi^2$) (*p*-Value) |
|---|---|---|---|---|---|
| | Yes N (%) | No N (%) | | | |
| Idutywa Clinic | 29 (24.8) | 88 (75.2) | 117 | | |
| Ngangalizwe Clinic | 74 (31.8) | 159 (68.2) | 233 | | |
| Nqamakwe Clinic | 47 (44.3) | 59 (55.7) | 106 | 3 | 17.99 (0.001) * |
| Tsolo Clinic | 30 (20.8) | 114 (79.2) | 144 | | |
| Total | 180 (30.0) | 420 (70.0) | 600 | | |

* Statistical significance ($p \leq 0.05$).

### 3.2.2. Prevalence by Gender

The rate of parasitic infection in females and males was 27.9% and 39.2%, respectively. The Chi-square test for independence was used to determine if a relationship existed between gender and the presence of a parasite. The results show $\chi^2 = 5.729$ (*p*-value = 0.017), implying a statistically significant higher prevalence of parasites in males than females. See Figure 2.

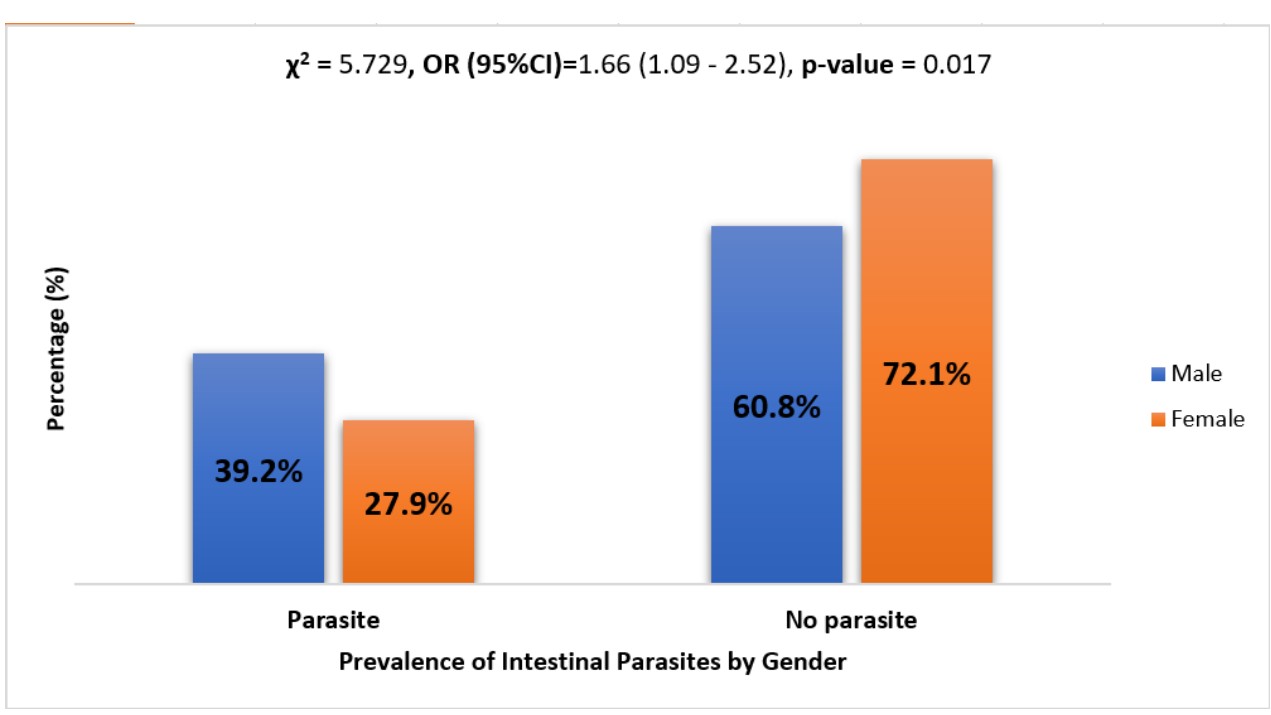

**Figure 2.** Prevalence of intestinal parasites by gender. Logistic regression analysis results show that males were 1.66 times more likely to have parasites as compared to their female counterparts (OR = 1.66, CI = (1.09–2.52), and *p*-value = 0.017).

### 3.2.3. Prevalence of Intestinal Parasites in Rural and Urban Dwellers

Prevalence of intestinal parasites according to place of residence of participants is shown in Table 3. The results show the prevalence of 30.7% for rural dwellers and 28.7% for urban dwellers. The results from the Chi-square test for independence show no statistically significant relationship between the prevalence of intestinal parasites and place of residence ($\chi^2 = 0.255$, df = 1, and *p*-value = 0.614).

**Table 3.** Prevalence of intestinal parasites by place of residence.

| Place of Residence | Intestinal Parasite | | Total | Df | Chi-Square ($\chi^2$) (*p*-Value) |
|---|---|---|---|---|---|
| | Yes N (%) | No N (%) | | | |
| Rural | 120 (30.7) | 271 (69.3) | 391 | | |
| Urban | 60 (28.7) | 149 (71.3) | 209 | 1 | 0.255 (0.614) |
| Total | 180 (30.0) | 420 (70.0) | 600 (100.0) | | |

3.2.4. Prevalence of Intestinal Parasites According to Age

Table 4 shows prevalence of intestinal parasites by age of participants. The study population cut across those between the ages of 12 and 75 years. The age group mostly infected with intestinal parasites was 12–20 (36.4%), followed by the age group 50–60 (32.1%). The Chi-square test for independence was used to determine if there was a relationship between age group and the presence of parasites. The results show $\chi^2 = 2.233$ (*p*-value = 0.897), implying no statistically significant association between the age group and the presence of parasites.

**Table 4.** Prevalence of intestinal parasites by age.

| Age | Intestinal Parasite | | Total | Df | Chi-Square ($\chi^2$) (*p*-Value) |
|---|---|---|---|---|---|
| | Yes N (%) | No N (%) | | | |
| 12–20 | 12 (36.4) | 21 (63.6) | 33 | | |
| 21–30 | 28 (26.4) | 78 (73.6) | 106 | | |
| 31–40 | 57 (31.5) | 124 (68.5) | 181 | | |
| 41–50 | 47 (29.0) | 115 (71.0) | 162 | 6 | 2.233 (0.897) |
| 51–60 | 25 (32.1) | 53 (67.9) | 78 | | |
| 61–70 | 10 (31.3) | 22 (68.8) | 32 | | |
| 71+ | 1 (16.7) | 5 (83.3) | 6 | | |
| Total | 180 (30.1) | 420 (69.9) | 598 | | |

3.2.5. Prevalence of Intestinal Parasites by the Type of Toilet Used

The highest prevalence of intestinal parasites was observed in traditional toilets (31.7%) as shown in Table 5. However, the results from the Chi-square test for independence show no statistically significant relationship between the prevalence of intestinal parasites and the type of toilet used ($\chi^2 = 3.944$, *p*-value = 0.268).

**Table 5.** Prevalence of intestinal parasites by type of toilet used.

| Type of Toilet Used | Intestinal Parasite | | Total | Df | Chi-Square ($\chi^2$) (*p*-Value) |
|---|---|---|---|---|---|
| | Yes N (%) | No N (%) | | | |
| Traditional | 128 (31.7) | 276 (68.3) | 33 | | |
| Modern | 48 (28.1) | 123 (71.9) | 106 | | |
| Both | 4 (20.0) | 16 (80.0) | 181 | 3 | 3.944 (0.268) |
| Bush | 0 (0.0) | 5 (100.0) | 162 | | |
| Total | 179 (30.1) | 417 (69.9) | 596 | | |

### 3.2.6. Distribution of Intestinal Parasites by Type of Toilet Used

The distribution of different types of intestinal parasites and percentage according to type of toilet used by participants are shown in Table 6. The result was tested with a Chi-square test among *Ascaris lumbricoides*, which had the largest distribution, followed by *Balantidium coli* and *Entamoeba coli*. The results show that among the infected persons, the majority uses traditional toilets compared to those using modern toilets. The Chi-square test for goodness of fit revealed a statistically significant increase in the number of *Ascaris* and *B. coli infections* observed in traditional toilets. The Chi-square test, however, showed no significant relationship between the number of people infected with *Entamoeba* and the other type of toilet.

**Table 6.** Distribution of intestinal parasites by type of toilet used.

| Type | Distribution of Intestinal Parasites | | | |
|---|---|---|---|---|
| | *Ascaris* | *B. coli* | *Entamoeba* | **Other** |
| Traditional | 80 (76.9) | 16 (57.1) | 13 (61.9) | 23 (69.7) |
| Modern | 21 (20.2) | 11 (39.3) | 8 (38.1) | 10 (30.3) |
| Both | 3 (2.9) | 1 (3.6) | - | 0 (0.0) |
| Total | 104 | 28 | 21 | 33 |
| Chi-Square ($\chi^2$) | 0.001 [μ,*] | 0.002 [μ,*] | 0.48 (0.668) | 1.26 (0.873) |

* Statistical significance ($p \leq 0.05$); μ = Fisher's exact $p$ (recommended where cell values are <5).

## 4. Discussion

The overall prevalence rate of intestinal infection detected in this study was 30%. A similar study that examined an overlap between HIV and helminth infections on nutritional value in KZN South Africa established a prevalence of 36.1% for HIV and helminths, respectively [11].

Another study conducted in Mthatha discovered a 25% prevalence of HIV [12]. In Baringo Kenya, the prevalence was 50.9% [9]. A more recent study in Northwest Ethiopia reported a prevalence of 45.3% in pre-ART patients and 20% in ART patients [14]. The prevalence of 30 percent detected in this study may be attributed to low socioeconomic status, unsanitary conditions, household clustering, and a lack of access to potable water. The prevalence at the Nqamakwe Clinic site was 44%. This community and its surroundings are located in Mnguma local municipality Butterworth, which has long struggled with water crises caused by the drying up of dams and breakdowns of trucks delivering water to communities. The then Eastern Cape Premier declared some of these districts a drought disaster area in 2015 [15]. The high prevalence found in this area suggests that patients may be underreporting their symptoms to ARV nurses as they were not treated for intestinal parasites, or that the nurses were unaware that intestinal parasites negatively impact HIV/AIDS patients and should be suspected when related symptoms are reported.

The most prevalent parasite detected was *Ascaris lumbricoides* 55.9%. The alarming rate of *Ascaris lumbricoides* detected in this study was similar but higher than the 42.0 % earlier detected in the same region [12]. An older study in KZN, South Africa, found *A.lumbricoides* as the most common helminth infections [16]. A high presence of *A.lumbricoides* could be due to household clustering suggesting that despite their reduced immunity, HIV/AIDS patients were still exposed to infectious eggs shed by families, which increased their vulnerability. Poor sanitary conditions associated with underdevelopment and lack of potable drinking water in most of communities increase the risks of intestinal parasites [13].

The opportunistic parasites detected by the study were *Balantidium coli* 15.1%, *Entamoeba coli* 11.3%, *Enterobius vermicularis* 1.1%, *Cryptosporidium* spp. 1.6%, and *Isospora belli* 0.5%. These have been associated with acute and chronic diarrhea and weight loss among those living with HIV/AIDS [17]. HIV infection has been shown to predispose the patient to intracellular opportunistic intestinal parasites, such as *Isospora*. Opportunistic

infection is considered one of the most frequent causes of hospitalization and death in HIV-infected individuals even in this ART era [18]. Diarrhea is common in HIV/AIDS patients and can lead to life-threatening complications, especially in those with CD4 counts below 200 cells [19]. *Isospora belli* is frequently associated with chronic diarrhea in developing countries, and it affects 5–26 percent of HIV/AIDS patients [20]. A study conducted in Australia on the presence of enteric parasites in HIV/AIDS patients discovered that *C. parvum* cases occurred exclusively in HIV patients [21].

Among the parasites detected was the *Taenia species* (3.8%). Mthatha has a high prevalence of neurocysticercosis, and it has taken a considerable monetary toll in the Eastern Cape region [22]. This study detected that *Taenia solium* is not just endemic in Mthatha but also occurred in three other sites outside Mthatha. This is not surprising since there is a high prevalence of *Taenia solium* taeniasis/cysticercosis in humans and pigs in the Eastern Cape Province (ECP) of South Africa [23]. *Taenia solium* is acquired in humans through the ingestion of the parasite's larval cysts in undercooked and infected pork. When they enter the central nervous system, *Taenia solium* causes an epileptic seizure [24]. It was observed that many *T. solium* regions in Sub-Saharan Africa are also endemic for the human immunodeficiency virus [25]. Multiple infections of *Ascaris* and other parasites were detected during the screening in five females and two males. The coexistences were observed between *Ascaris* and *Balantidium coli*, *Ascaris* and *F. busiki*, *Ascaris* and *Entamoeba coli*, *Ascaris* and *Schistosoma mansoni*, and *Ascaris* and *D. latum.* There was also a coexistence of *Taenia solium* and *Entamoeba coli*. Multiple infections are a major influencer on the evolution of infectious diseases and, specifically, on their virulence [26]. Multiple infections tend to thrive in clustered, poor environments where there is inadequate sanitation. While screening for intestinal parasites among HIV patients, it was found that participants were predominantly female 79.9% and male 22.1% (476/120). This agreed with a cross-sectional report on intestinal parasites among HIV patients report by Sokoto, which had predominantly female attendants (64.9%) [27]. The fecal samples collected from Brazil also found that 54.44% of the participants were female, while 45.55% were male [28]. Despite the high participating number of females, the prevalence of intestinal parasite was higher in the fewer males than the females: 39.2% and 27.9%, respectively. These findings are consistent with a study in Nigeria where intestinal parasitic infection was generally higher in males (57.60%) than females (42.40%) [29]. The preponderance of high female HIV/AIDS clinic participants could elaborate the findings that women are tested for HIV earlier—often in an antenatal setting and are put on ART earlier [30,31]. It could also mean that women are grossly affected by HIV/AIDS because some were subjected to early marriage and sex with or without protection, which affects their health, education, and economic status [32].

## 5. Conclusions

The high prevalence of intestinal parasites found in both rural and urban patients demonstrates that patients with HIV/AIDS, despite receiving antiretroviral therapy, continue to face complications from both infections. As a result, a holistic approach that includes periodic screening and deworming as well as sanitation is required to eradicate parasites and provide them with optimal health and a high quality of life.

**Author Contributions:** Conceptualization, D.T.A.; Methodology, D.T.A., T.A.; Data collection, A.I., B.A.; Manuscript development, A.I., D.T.A.; Manuscript revision, D.T.A., O.O.; Correspondence author, D.T.A.; Manuscript submission, O.O. All authors have read and agreed to the published version of the manuscript.

**Funding:** This research received no external funding.

**Institutional Review Board Statement:** This research was conducted in accordance with the Declaration of Helsinki and was approved by faculty of health sciences postgraduate education, training, research and ethics unit, and human research committee of Walter Sisulu University.

**Data Availability Statement:** The dataset for this study is available from the corresponding author upon request and with rightful permission from the affiliated institution.

**Conflicts of Interest:** The authors declare no conflict of interest.

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
