# Peer review of "Prevalence of Intestinal Parasites in HIV/AIDS-Infected Patients Attending Clinics in Selected Areas of the Eastern Cape"

_2036-7481, doi:10.3390/microbiolres13030040_

Round 1
Reviewer 1 Report
Authors deal with a very important topic, which needs an attention.
I have only a few suggestions:
61-71 2.1 Study Setting – repeated reference source (13)
100 Isospera – should be Isospora, Cyclospora cayentanesis – should be Cyclospora cayentanensis
101-102 The duration of the study was from March 2019 to February 2020. The samples from the four sites were collected and analyzed between March 2019 and February 2020. - Is it necessary to explicitly state the duration of the study when it is the same as the collecting interval?
141-147 The parasites at the study were Ascaris lumbricoides 55.9%, Balantidium coli 15.1%i, Entamoeba coli 11.3%, Diphyllobothrium latum 4.3%, Taenia species 3.8%, Entamoeba Ver- mucularis 3.8%, Schistosoma mansoni 2.1%, Cryptosporidium Parvum 2.0%, Fasciolopsis bus- iki1.1%, Trichuris trichiuria1.1%, isospera belli 0.5%, Hymenolepsis nana 1.6%, Fasciola hepat- ica 0.5%, Trichostrongylus specie 0.5%. The opportunistic parasites found among them were cryptosporidium parvum, Entamoeba coli, Balantidium Coli, Isospera belli, entamoeba Ver- mucularis and they accounted for 30.5%. The most detected helminth was Ascaris (55.9 %) and most detected protozoan was Balantidium coli (15.1 %).
- Here and further in the text there are many typos considering the latin names and the use of zoological nomenclature (italic, capital letters, abbreviations). It is necessary to correct this in the whole manuscript.
E. g. in this paragraph it should be like this:
The parasites at the study were Ascaris lumbricoides 55.9%, Balantidium coli 15.1%, Entamoeba coli 11.3%, Diphyllobothrium latum 4.3%, Taenia spp. 3.8%, Entamoeba vermicularis 3.8%, Schistosoma mansoni 2.1%, Cryptosporidium parvum 2.0%, Fasciolopsis buski 1.1%, Trichuris trichiura 1.1%, Isospora belli 0.5%, Hymenolepis nana 1.6%, Fasciola hepatica 0.5%, Trichostrongylus spp. 0.5%. The opportunistic parasites found among them were Cryptosporidium parvum, Entamoeba coli, Balantidium coli, Isospora belli, Entamoeba vermicularis and they accounted for 30.5%. The most detected helminth was Ascaris lumbricoides (55.9 %) and most detected protozoan was Balantidium coli (15.1 %).
If the parasites’ names are used repeatedly, it is sufficient to use the full name only for the first time, then you can use abbreviation, e. g.:- Balantidium coli -> B. coli.
In Table 1. There are many Socio-demographic characteristics, however, not all of them are discussed and statistically evaluated further in the text regarding the prevalence of parasites. I suppose it would be better either not to mention the undiscussed characteristics or discuss all of them (in the same order as they are stated in the Table1).
Author Response
|
Comment |
Response |
Page; line; paragraph |
Remarks |
|
1-71 2.1 Study Setting – repeated reference source (13) |
Repeated phrase deleted |
2&3; 35-39; 3 |
|
|
100 Isospera – should be Isospora, Cyclospora cayentanesis – should be Cyclospora cayentanensis
|
Isospera corrected to Isospora cayentanesis corrected to cayentanensis |
3; 28; 4 |
|
|
101-102 The duration of the study was from March 2019 to February 2020. The samples from the four sites were collected and analyzed between March 2019 and February 2020. - Is it necessary to explicitly state the duration of the study when it is the same as the collecting interval?
|
This section is restructured based on how the sampling procedure was done. Explicit explanation is given under the section: “Sampling procedure” |
3; 1-6; 8 |
Authors appreciate reviewer’s advice and implement same as advised. However, authors considered the duration of sampling procedure more pertinent, given its relevance to the manuscript. The Study was for a degree purpose and has a three year duration. |
|
141-147….here and further in the text there are many typos considering the latin names and the use of zoological nomenclature (italic, capital letters, abbreviations). It is necessary to correct this in the whole manuscript |
All typographical errors, including correct zoological nomenclatures were corrected |
6; 1-9; 1 |
|
|
If the parasites’ names are used repeatedly, it is sufficient to use the full name only for the first time, then you can use abbreviation, e. g.:- Balantidium coli -> B. coli.
|
Corrected |
6; 1-9; 1 |
|
|
In Table 1. There are many Socio-demographic characteristics, however, not all of them are discussed and statistically evaluated further in the text regarding the prevalence of parasites. I suppose it would be better either not to mention the undiscussed characteristics or discuss all of them (in the same order as they are stated in the Table1).
|
Table 1 presents demographic variables with introductory narratives. However, only variables considered to be statistically significant were further discussed down the following paragraphs. |
|
|

Reviewer 2 Report
The work is interesting but it should be improved and particularly a lot of mistakes concerning parasites and English
Species names...in italic font
Taenia spp. and not Taenia sp
There are different size font in the text, change
Isospora belli not Isospera
Males were affected more frequently (27.9%) than females (39.2%)....??????
p-value or P-value...change following guidelines of journal
entamoeba Vermucularis 1.1 %???????
Cryptosporidium Parvum...change in Cryptosporidium parvum
Enterobius vermicularis...Didd the authors analyzed the prevalence of this parasite???they used scotch test to perform analysis??
other numerous mistakes...
Author Response
|
Comment |
Response |
Page; line; paragraph |
Remarks |
|
The work is interesting but it should be improved and particularly a lot of mistakes concerning parasites and English
|
Whole manuscript improved on spellings, typographical errors and grammar. Language editor edited the whole manuscript |
Whole manuscript |
Authors appreciate this very comment and suggestion immensely |
|
Species names...in italic font
|
All species names that were not italicized are revised and written in italics |
Whole manuscript |
|
|
Taenia spp. and not Taenia sp
|
Corrected |
6;5;1 |
|
|
There is different size font in the text, change
|
Uniform font size (calibri body, 12) effected |
Whole manuscript |
|
|
Isospora belli not Isospera
|
Isospera corrected to Isospora |
3; 28; 4 |
|
|
Males were affected more frequently (27.9%) than females (39.2%)....??????
|
The correct position here, as seen from the findings of this study is that, the percentage of male that were found to be parasitized is higher (39.2%) compared to female (27.9%) |
8; 1; 1 |
Authors highly regret this simple but costly mistake in results’ presentation |
|
p-value or P-value...change following guidelines of journal
|
p-value upheld |
Throughout the manuscript |
|
|
entamoeba Vermucularis 1.1 %???????
|
The correct parasite in reference is Enterobius vermicularis, and not entamoeba Vermucularis |
|
Authors highly regret this misinformation |
|
Cryptosporidium Parvum...change in Cryptosporidium parvum
|
Correction made |
Everywhere in the manuscript |
|
|
Enterobius vermicularis...Didd the authors analyzed the prevalence of this parasite???they used scotch test to perform analysis??
|
|
|
|
|
other numerous mistakes...
|
Whole manuscript edited and revised to meet the guidelines of the journal |
|
|

Round 2
Reviewer 2 Report
The work was improved but there are several errors yet, particularly species name, italics name...they must be corrected!!!!
page 4---lane 155-158
page 8---lane 260-263
page 8---lane
page 9---lane 276
Figure 1...a lot of errors
Author Response
25 July 2022
The Editor,
We thank the Handling Editor and reviewers for their efforts. We are pleased that there is interest in this topic and are grateful for the constructive comments that have improved the manuscript.
We have addressed each of the reviewers’ comments in a point-by-point manner in the table below. While hoping that these changes will meet with your favorable consideration, we hold ourselves at your disposition for any further clarifications.
|
S/N |
Reviewer comment |
Revision |
|
1. |
Diarrhoea is very common in HIV/AIDS patients. Comment on potential associations between diarrhoea and the presence of a given intestinal parasite species. |
This comment was expanded as follows: Diarrhoea is very common in HIV/AIDS patients and can cause life threatening complications, particularly in HIV/AIDS patients with CD4 counts <200 cells (19). In developing countries, Isospora belli is frequently associated with chronic diarrhea and it occurs in 5-26% among HIV/AIDS patients (20). A study in Australia, which examined the presence of enteric parasites in HIV/AIDS, showed that C. parvum cases occurred exclusively in patients with HIV (21). |
|
2. |
Microsporidia (e.g. Enterocytozoon bieneusi, Encephalitozoon spp.) are common opportunistic pathogens. |
This has been placed in the appropriate space. Microsporidia such as Enterocytozoon bieneusi, Encephalitozoon spp were mentioned |
|
3. |
Cryptosporidium parvum |
Replaced with Cryptosporidium spp |
|
4. |
page 4---lane 155-158 |
This part has now been revised |
|
5. |
page 8---lane 260-263
|
Additional information on diarrhea were made. Changes in italics, grammar, nomenclatures |
|
6. |
page 8---lane
|
Additional information on diarrhea were made. Changes in italics, grammar, spellings nomenclatures |
|
7. |
page 9---lane 276 |
Reversed grammar |
|
8. |
Figure 1...a lot of errors |
This has now been presented correctly |
Yours’
Olanrewaju Oladimeji for all authors
